# *MAM-E*: Mammographic Synthetic Image Generation with Diffusion Models

**DOI:** 10.3390/s24072076

**Published:** 2024-03-24

**Authors:** Ricardo Montoya-del-Angel, Karla Sam-Millan, Joan C. Vilanova, Robert Martí

**Affiliations:** 1Computer Vision and Robotics Institute (ViCOROB), University of Girona, 17004 Girona, Spain; sammillan95@gmail.com (K.S.-M.); robert.marti@udg.edu (R.M.); 2Department of Radiology, Clínica Girona, Institute of Diagnostic Imaging (IDI) Girona, University of Girona, 17004 Girona, Spain; joancarles.vilanova@udg.edu

**Keywords:** diffusion models, mammography, lesion inpainting

## Abstract

Generative models are used as an alternative data augmentation technique to alleviate the data scarcity problem faced in the medical imaging field. Diffusion models have gathered special attention due to their innovative generation approach, the high quality of the generated images, and their relatively less complex training process compared with Generative Adversarial Networks. Still, the implementation of such models in the medical domain remains at an early stage. In this work, we propose exploring the use of diffusion models for the generation of high-quality, full-field digital mammograms using state-of-the-art conditional diffusion pipelines. Additionally, we propose using stable diffusion models for the inpainting of synthetic mass-like lesions on healthy mammograms. We introduce *MAM-E*, a pipeline of generative models for high-quality mammography synthesis controlled by a text prompt and capable of generating synthetic mass-like lesions on specific regions of the breast. Finally, we provide quantitative and qualitative assessment of the generated images and easy-to-use graphical user interfaces for mammography synthesis.

## 1. Introduction

Data scarcity is an important problem faced in the medical imaging domain, caused by several factors, such as expensive image acquisition, processing and labeling procedures, data privacy concerns, and the rare incidence of some pathologies [1]. This leads to a reduction in the volume of medical data available for the training of deep-learning models, which limits the models’ performance and holds back the development of computer-aided systems, compared with non-medical imaging applications.

Generative models have been used to complement traditional data augmentation techniques and expand medical datasets, with generative adversarial networks (GANs) being, for many years, the state-of-the-art (SOTA) due to their high image quality and photorealism. Nevertheless, unstable training, low diversity generation, and low sample quality make the use of GAN-like architectures challenging for medical imaging synthesis [1]. Because medical diagnosis can depend on subtle changes in organ appearance reflected in the images, realistic high-quality synthetic generation is crucial for the reliable performance of computer-assisted diagnosis and intervention systems [2].

Diffusion models (DM) captured special attention from the generative models community when they were proposed by Dhariwal and Nichol [3] for image generation and seemingly outperformed GANs in 2021. Since then, applications and research papers for medical images have been published to explore this new image generation principle. For instance, Dorjsembe et al. [4] proposed using the original pipeline of diffusion models on computer vision, called denoising diffusion probabilistic models (DDPM) [5], for the generation of high-quality MRI of brain tumors. This first implementation of diffusion models for 3D medical images reached SOTA results and outperformed the baseline models based on 3D GANs. Further advances in the field led to latent diffusion [6], which introduces the use of a latent space for higher image resolution. Latent diffusion was used by Pinaya et al. [7] to generate high-resolution 3D brain images, increasing the image resolution from 64 × 64 × 64 to 160 × 224 × 160 without requiring more GPU memory usage or overall training time. The Fréchet inception distance (FID) [8] for image fidelity and the multi-scale structural similarity index measure (MS-SSIM) for generation diversity were computed, and in both cases DM surpassed the GANs’ baseline metrics.

A more controlled generation process can be achieved by feeding additional input during training and inference. An example of this is stable diffusion (SD) [6], a conditional diffusion model with text prompts as generation conditioning. An SD implementation for medical images was introduced by Chambon et al. [9], who proposed a model for chest X-ray generation. Their model, named *RoentGen*, was able to create visually convincing, diverse chest X-rays, controlling the generation output using text prompts with radiology-specific language. A key characteristic of this work is the use of SD weights pretrained with natural images as the baseline. Instead of training from scratch, specific parts of the network are fine-tuned to adapt the weights from its original domain into a specific chest X-rays medical domain. This DM fine-tuning approach is called *Dreambooth* and was first introduced by Ruiz et al. [10] for natural images.

Besides image generation, DM can be used for other tasks, such as super-resolution, image denoising, and inpainting. Hung et al. [11] used a conditional diffusion model to inpaint MRI prostate images to recover their original information, outperforming other generation methods in both qualitative visual inspection and quantitative generation metric comparison. Some works have explored lesion inpainting using DM for brain MRI. Rouzrokh et al. [12] developed a DDPM to execute several inpainting tasks, such as generating synthetic lesions or healthy tissue on slices of the 3D MRI volumes in various sequences. Their model was capable of generating realistic tumoral lesions and tumor-free brain tissue, although the performance of the model was only assessed visually.

### 1.1. Generative Models for Mammography

In the specific case of mammography, the generation of synthetic images can be used to improve the performance of computer-aided detection and diagnosis (CADe and CADx) systems, the main artificial intelligence techniques used to assist clinical decision-making [13]. Classification tasks such as breast density classification [14,15] and immunohistochemical status classification [16,17] are examples of mammography-based CAD systems suffering from an imbalanced dataset problem. Other applications, such as lesion classification, could benefit from the generation of specific objects, such as synthetic lesions inpainting to augment the sample size. Some works have explored the use of GANs to generate synthetic images, such as [18,19] for full-field mammograms generation and [20] for lesion generation. Their result show that although GANs are a viable technique for image generation, they suffer from important limitations, such as high-resolution scalability limits, mode collapse, and difficulty training the adversarial models.

Diffusion models are being presented as a plausible alternative for data augmentation, and their implementation may complement other data augmentation techniques. Nevertheless, the use of diffusion models in the medical imaging field continues at early stages, specially for mammography. Prior to this publication, we released the source code, weights, and user interface for the first implementation of SD for mammographic image synthesis in the author’s personal GitHub repository [21]. Following works have explored the generation of synthetic mammograms using DM, such as the release of one synthetic mammography dataset from Pinaya et al. [22], composed of 100 k 512 × 512 synthetic images with masking level labeling, and the proposal of Kidder [23] to explore the use of SD for contrast-enhanced spectral mammography generation.

### 1.2. Our Proposal

In this paper, we introduce *MAM-E*, a pipeline of diffusion models for high-quality mammographic image synthesis, capable of generating images based on a text prompt description, and also capable of generating lesions on a specific section of the breast using a mask. Our pipeline was developed using stable diffusion, a SOTA diffusion model technique that leverages both conditioning, to control the image generation, and a latent space to allow high resolution without requiring large computational resources. The generated images are *for presentation*, meaning that their appearance and pixel intensities are meant for radiologist inspection, with the limitations on resolution and pixel depth inherent to the diffusion pipelines. Our main workflow can be separated into two tasks: healthy mammogram generation and lesion inpainting. For the first task, the generation process is controlled by text conditioning, with a description of the image including view, breast density, breast area, and vendor. For the second task, we use a stable diffusion inpainting model designed to generate synthetic mass-like lesions in desired regions of the mammogram. The name of our model was inspired by OpenAI’s DALL-E image generation AI system [24].

The main contributions of this paper are the following: (1) To the knowledge of the authors, this is the first work that offers customized mammographic image generation using text-conditioning stable diffusion, which can be used by both computer scientists for the improvement and optimization of AI algorithms, and by radiologists and medical practitioners for educational proposes. (2) This is also the first work that implements stable diffusion for lesion inpainting for mammography. (3) The combination of both tasks into a single pipeline allows the generation of completely synthetic mammograms with or without mass-like lesion. (4) Through the combination of two datasets of different populations, we discovered that concepts such as breast density and area can be shared among datasets, allowing the generation of mammograms with labels not included in their original individual datasets, a phenomenon we called *concept extrapolation* (See Section 3.1.2).

This work source code publication is the first implementation of SD for mammographic image generation and can be found at https://github.com/VICTORIA-project/mam-e (accessed on 21 March 2024), along with the pretrained weights at https://huggingface.co/Likalto4 (accessed on 21 March 2024). Additionally, graphical user interfaces for both synthesis tasks were designed, as shown in Figure 1, for easy-to-use image generation.

## 2. Materials and Methods

### 2.1. Datasets

We decided to use two datasets for the training of the diffusion models so that different patient populations and mammography unit vendors were considered. A summary of the distribution of the cases can be found in Table 1.

#### 2.1.1. OMI-H

We used a subset of the OPTIMAM Mammography Image Database, consisting of approximately 40 k Hologic full-field digital mammograms (FFDM) from several UK breast-screening centers and with different image views [25]. The dataset was composed of mammograms with and without lesions (masses of type benign, malignant, and interval-cancers), and expert annotations are included in the respective cases, including the coordinates of a bounding box surrounding the lesion. The images were in CC and MLO views and no breast implant was present in any image.

#### 2.1.2. VinDr-Mammo

A second dataset composed of approximately 20 k FFDM with breast-level assessment and extensive lesion annotation was also used. It consists of 5000 mammography exams, each with 4 standard views (CC and MLO for both lateralities), coming from two primary hospitals from Vietnam, giving a total of 20,000 images in DICOM files [26]. The metadata of each image, consisting of both technical and clinical information, was also available in a CSV file. We filtered the images so that only mammograms coming from a Siemens vendor unit were used, obtaining a final set of 15,475 mammograms. The lesions present in the images correspond to masses and in some cases masses with suspicious microcalcifications.

### 2.2. Data Preprocessing and Preparation

Both datasets were subject to the same preprocessing and preparation steps. First, mammograms were saved as PNG files to ensure faster access and less disk memory space. Secondly, to be able to use pretrained weights, the images were saved in RGB format, repeating the original gray-channel into each RGB channel. The original image intensities with uint16 data types were scaled to a [0, 255] range with a reduced uint8 data type.

Additionally, in order to use the pretrained weights available for SD, the images were resized to a 512 × 512 square using bilineal interpolation and center cropping, as shown in Figure 2. Images with right laterality were horizontally flipped so that all images had the breast region on the same side.

#### 2.2.1. Healthy Image Generation

For each healthy mammogram, a text prompt description was created and saved along with the image ID in a JSON file. In the case of the OMI-H dataset, we created a prompt with the image view and breast area size information. To compute the breast area size, we first obtained a breast mask using the intensity information of the image and then applied a threshold to separate background and breast tissue. Later, the ratio of pixels in the mask compared with the total image was computed and criteria for three different breast area sizes was defined, as shown in Table 2. For the VinDr dataset, the breast density information was included instead of the breast area for the prompt description. Breast density was available in BI-RADS scale, so we needed to transform this information into a semantically meaningful text following the criteria in Table 2, based on the BI-RADS breast density descriptions [27]. Examples of the text–image training samples can be found in Figure 3.

#### 2.2.2. Lesion Inpainting

The inpainting task requires mammograms with confirmed lesions only. Using the bounding boxes coordinates available in the metadata, binary masks were generated. Naturally, due to the resizing and cropping preprocessing performed previously, the original coordinates required a proper redefinition using simple geometrical properties. The mask has pixel values of 255 inside the bounding box and zero elsewhere. Because the SD architecture used for the inpainting task requires an input text prompt for the generation, a toy prompt with “a mammogram with a lesion” text was used for all training images.

### 2.3. Diffusion Models

The original diffusion model idea was presented by Sohl-Dickstein et al. [28] and consisted of using a Markov chain, a sequence of stochastic events whose time steps depend on the previous one, to gradually convert one known distribution (e.g., Gaussian distribution) into another (target distribution). Inspired by non-equilibrium statistical physics, the main idea is to systematically and iteratively destroy structure in a data distribution through a process called forward diffusion. Then, the reverse diffusion process is learned and used to restore structure in the data. The first practical implementation of the DM premise on images was developed by Ho et al. [5] introducing *Denoising diffusion probabilistic models* (DDPM). In this framework, the data are destroyed by adding Gaussian noise to the image in an iterative fashion described by the Markov chain, as seen in Figure 4. The total number of diffusion timesteps, *T*, is decided by the user, but a usual number is around T=1000. To learn the reverse process, a UNet is used to carry on the denoising process. As such, a DM is built from three fundamental components: a noise scheduler to add noise in the forward process, a UNet to denoise in the reverse process, and a timestep encoder to encode the timestep, *t*, into the UNet. More information regarding each component is provided in the following sections.

#### 2.3.1. Forward Diffusion Process

Let $x_{0}$ be the original image and $x_{t}$ the noisy version of that image at time *t*. For the forward diffusion, we can define the Markov chain process as
(1)$$q\left(x_{1:T}|x_{0}\right)=\Pi_{t=1}^{T}q\left(x_{t}|x_{t-1}\right),$$
where *q* is a probability distribution from which the noisy version of the image at time *t* can be sampled, given $x_{t-1}$. The proposal of the DDPM framework [5] is to define *q* as a Gaussian (normal) distribution given by
(2)$$q\left(x_{t}|x_{t-1}\right)=N\left(x_{t};\sqrt{1-\beta_{t}}x_{t-1},\beta_{t}I\right),$$
where $x_{t}$ is the output of the distribution sampling, $\sqrt{1-\beta_{t}}x_{t-1}$ is the mean, and β is the variance of the distribution. Therefore, the sampling of the next noisy version of the image is essentially controlled by β, as its value affects both the mean and the variance of the sampling distribution. Selecting the manner in which β changes through time is called beta scheduling and is control by the noise scheduler. In Figure 5a,c two examples of beta scheduling are shown.

Thanks to the additive properties of Gaussian distributions, we can obtain a noisy image at any timestep, *t*, directly by rewriting the sampling distribution Equation (Equation 2) as
(3)$$q\left(x_{t}|x_{0}\right)=N\left(x_{t};\sqrt{{\tilde{\alpha }}_{t}}x_{0},\left(1-\tilde{\alpha }\right)I\right),$$
with ${\tilde{\alpha }}_{t}=\Pi_{s=1}^{t}\alpha_{s}$ and $\alpha_{t}=1-\beta_{t}$, where α can be interpreted, as a measure of much information from the previous image is being kept during the diffusion process. The importance of ${\tilde{\alpha }}_{t}$, and therefore of $\beta_{t}$, can be understood by looking at Figure 5. For *t* values close to 0, the distribution from which we sample has μ≈1 and σ≈0, meaning that the sample images are every similar to the original image. On the other hand, for large *t* values where μ≈0 and σ≈1 the distribution is close to a standard normal distribution (SND) and the sampled image will be essentially pure Gaussian noise.

Finally, to be able to define the training goal in the reverse diffusion process, we express the sampling from the probability distribution in Equation (Equation 3) using the *reparameterization trick* [29]. The reparameterization trick allows us to write the generation of a sample *X* from a normal distribution N(μ,σ) as X=μ+σZ, where Z∼N(0,1), i.e., *Z* was sampled from an SND. With this, the forward diffusion sampling process can be expressed by
(4)$$x_{t}=\sqrt{{\tilde{\alpha }}_{t}}x_{0}+\sqrt{1-{\tilde{\alpha }}_{t}}\epsilon ,$$
where ϵ∼N(0,1). The stochastic variable epsilon (ϵ) in Equation (Equation 4) is crucial to understand the reverse diffusion process as it is essentially the prediction target of the UNet.

#### 2.3.2. Reverse Diffusion Process

The reconstruction of the data destroyed by noise can be performed using a UNet that has learned to denoise the images. Formally, the reverse process is also a Markov chain that can be defined in a similar way as
(5)$$p_{\theta }\left(x_{0:T}\right)=p\left(x_{T}\right)\Pi_{t=1}^{T}p_{\theta }\left(x_{t-1}|x_{t}\right),$$
where $p_{\theta }$ is the learned probability distribution from which the denoised images are sampled at each timestep, *t*. θ indicates that the distribution is parameterized as it was learned by the UNet. This also explains why the term $p(x_{T})$ has no subscript θ as it is the starting point of the reverse process, i.e., pure Gaussian noise.

Assuming that *p* can also be modeled as a normal distribution, it can be expressed as
(6)$$p_{\theta }\left(x_{t-1}|x_{t}\right)=N\left(x_{t-1};\mu_{\theta }\left(x_{t},t\right),\Sigma_{\theta }\left(x_{t},t\right)\right),$$
where $\mu_{\theta }$ and $\Sigma_{\theta }$ are the learnable mean and variance of the reverse sampling distribution. To reduce the training complexity, and because it has been shown to provide similar results, $\Sigma_{\theta }=\beta I$; therefore, only $\mu_{\theta }$ has to be learned. Due to limitations of space, the complete formulation of the optimization of the usual variational bound on negative log likelihood is not fully described, but key considerations of this formulation are given instead. The first consideration is that $\mu_{\theta }$ can be computed as
(7)$$\mu_{\theta }\left(x_{t},t\right)=\frac{1}{\sqrt{\alpha_{T}}}\left(x_{t}-\frac{\beta }{\sqrt{1-{\tilde{\alpha }}_{t}}}\epsilon_{\theta }\left(x_{t},t\right)\right),$$
where the key is to notice that only $\epsilon_{\theta }$ is needed to predict $\mu_{\theta }$.

The second consideration is that the optimization term, which consequently defines the loss function of the UNet, is
(8)$$L=∥\epsilon -\epsilon_{\theta }{∥}^{2},$$
where epsilon (ϵ) is the same Gaussian noise defined in Equation (Equation 4), sampled from a Gaussian distribution ϵ∼N(0,1), and $\epsilon_{\theta }$ is the output of the UNet. In other words, the UNet objective is to implicitly learn the data distribution by predicting the scaled Gaussian noise, ϵ, added to the images at timestep *t*. To include the timestep as an additional input to the Unet, a timestep encoder is used to embed this information and use it during training.

Figure 6 shows the components of the denoising UNet used during the reverse process. It should be noted that the output of the UNet is the noise, $\epsilon_{0}$, that must be removed from the input image to approximately recover the original image without noise.

#### 2.3.3. Latent and Stable Diffusion

To solve the image size limitation, latent diffusion was introduced by Rombach et al. [6], using encoders to compress images from their original sizes in the image space into a smaller representation in the latent space. The motivation behind this is that images usually contain redundant information and an encoder can produce a smaller representation that can later be reconstructed back using a decoder. Therefore, in latent diffusion the diffusion process is performed on the latent representations rather than on the original images.

Stable diffusion is an improvement to latent diffusion, in which text conditioning is added to the model for additional control on the generation process [6]. The text conditioning is a prompt with the description of the image. To create a numeric representation of the prompt, a pretrained transformer called CLIP is used [30]. CLIP, which stands for Contrastive Language-Image Pretraining, maps both text and images into the same representational space, allowing comparison and similarity quantification between them [31].

#### 2.3.4. Fine-Tuning SD: DreamBooth

In 2022, Stability AI and LAION made the pretrained weights of their stable diffusion models [6] publicly available, which allowed the GM community to train domain-specific, fine-tuned SD models. Nevertheless, fine-tuning a large text-to-image model and teaching it new concepts can be challenging as difficulties such as catastrophic forgetting, overfitting, and low image generation diversity may emerge.

Ruiz et al. [10] presented an approach for fine-tuning SD called *DreamBooth*. They proposed using only a few images of the new subject with its respective text prompt to train the model using a small learning rate. Additionally, if the subject semantically exists in the model domain, prior generation images can be included for the training. This allows the binding of the new subject to a new unique identifier in the text-embedding space, as well as a learned representation in the pretrained data distribution. This fine-tuning technique has been tried for chest X-rays by Chambon et al. [9] and showed promising results on adapting the SD domain into their images to generate high-fidelity and diverse images thanks to the control given by the text prompt.

#### 2.3.5. Inference: Image Generation

With the UNet prediction, we can denoise pure Gaussian noise and generate new mammograms. The procedure is as follows:Sample random Gaussian noise ϵ∼N(0,I)for t=T,…,1 do:z∼N(0,I) if t>1 else z=0$x_{t-1}=\frac{1}{\sqrt{\alpha_{T}}}\left(x_{t}-\frac{\beta }{\sqrt{1-{\tilde{\alpha }}_{t}}}\epsilon_{\theta }\left(x_{t},t\right)\right)+\sigma_{t}z$end forDecode image using VAE

First, random Gaussian noise is sampled as a starting point. Then, the denoising process is repeated for *T* steps. The loop consists of using the predicted noise, $\epsilon_{\theta }$, to compute the distribution mean using Equation (Equation 7). By adding $\sigma_{t}z$ to this mean term, we are essentially sampling from the learned data distribution of the reverse diffusion process. After the denoising process is finished, the image is send back to the image space using the VAE decoder.

The inference process has two main hyperparameters to consider: number of timesteps, *T*, and the guidance scale. First, the number of timesteps, *T*, will depend on the type of sampling method that we use for denoising. The traditional DDPM sampling requires approximately 100 steps to generate good quality images, which is time consuming and represents a bottleneck in the image generation. The best alternative we found was to use the DPM-solver proposed by Lu et al. [32], which allows fast diffusion sampling with only 20 steps for good-quality image generation. In the result section, we show how the change of *T* affects the image quality.

The second hyperparameter is called the guidance scale. Even though the SD architecture uses cross attention in several parts of the network, so that the generation process focuses on the text prompt, in practice the model tends to ignore the text prompt at inference time. To solve this issue, Ho and Salimans [33] proposed a technique called classifier-free guidance. In essence, classifier-free guidance consists of generating two noise predictions, ϵ, at each step, one using the prompt ($\epsilon_{text}$) and one without it ($\epsilon_{free}$). Then, the difference between the prompt-generated noise and the free-generated noise is computed. This difference can be considered as a vector in the image distribution space, which points in the direction of the image with text. As such, we can scale this vector and sum it to the free-generated noise to force it to go more in the direction of the prompt text. This geometrical trick is illustrated in Figure 7.

Formally, the scaling factor is called guidance scale, and the formulation can be summarized as follows:(9)$$\epsilon_{\theta }=\epsilon_{free}+guidance*\left(\epsilon_{text}-\epsilon_{free}\right).$$

### 2.4. Implementation Details

Our experiments were conducted using stable diffusion models for both generation tasks, adapting the DreamBooth fine-tuning technique with pretrained *stable-diffusion-v1-5* weights as the baseline, publicly available in the *Hugging Face* model hub repository [6].

Figure 8 provides an overview of the full MAM-E pipeline proposed, which includes both image generation and lesion inpainting tasks. The following sections explain the specific details of each generation task.

#### 2.4.1. Latent Space Encoding

We decided not to fine-tune the VAE encoder and decoder after testing its encoding–decoding performance on our mammograms using pretrained natural image weights, as shown in Figure 9. Moreover, Chambon et al. [34] found that a pretrained VAE on natural images can perform well on Chest X-ray images. Using this VAE encoder, an original image of 512 × 512 pixels can be compressed to 4 latent representations of 64 × 64, reducing 16 times its original shape [29]. Consequently, the diffusion process is performed on the latent representations rather than on the original images, allowing lower memory usage, fewer layers in the UNet, and faster training and generation.

#### 2.4.2. Healthy Image Generation

For each dataset, we trained a separate model using only healthy images, as the datasets contain independent semantic information in the prompt and because the intensity ranges and image details differ between populations. Additionally, a third model with the combination of mammograms from both datasets was trained, adding the vendor’s name to the prompt.

Freezing the VAE weights, only the CLIP text encoder and the UNet weights were trained. The UNet architecture was the original stable diffusion denoising UNet proposed by [6] and can be seen in Figure 10. The network has four down- and upsampling blocks. Except for the last downsampling block (and its corresponding upsampling block), all blocks are composed of two ResNet blocks and two transformer blocks, one after the other. The timestep embedding is added to the first-layer ResNet blocks, whereas the text embeddings are added through cross attention into the Transformer blocks of all layers. For the last downblock (and first upblock), only the timestep information is fed.

The main training hyperparameters explored were the following:Batch size: 8, 16, 32, 64, 128, and 256.Training steps: Experiments ranged from 1 k up to 16 k.Learning rate: Three main values $1\times 10^{-6}$, $1\times 10^{-5}$, $1\times 10^{-4}$.

To select the best hyperparameters and to track the performance of the models, a validation process was conducted by generating 4 sample images from the same random Gaussian noise every 100 or 200 training steps. The training loss (mean squared error) and the GPU memory usage were also logged.

#### 2.4.3. Lesion Inpainting

The SD pipeline described for healthy mammogram generation can be modified in some key aspects to perform the inpainting of realistic mass-like lesions on mammograms. We propose using the modified DreamBooth fine-tuning pipeline to inpaint a lesion in a designated region of the breast. This pipeline was originally proposed for the inpainting of natural images in the *Hugging Face* hub to complete missing parts of images with specific text-prompted objects.

At training time, for each mammogram with the presence of a lesion two new elements are added per example: the mask and a masked version of the original image. The masked version of the original image is a copy of it where the pixel values inside the bounding box are set to zero. At training time, both the image and the masked image are first encoded into the latent space using the VAE encoder. Then, the mask is reshaped to match the latent representation size of the images. The text prompt and the timesteps were given similarly to the full-field image generation training pipeline. The training of the reverse diffusion process remains the same, except for one crucial difference: instead of feeding only the latent representation to the UNet, the latent representation, the mask, and the masked latent representation are stacked into one tensor and fed into the UNet, as seen in Figure 11. This small change in the training process allows the network to pay attention only to the pixels inside the mask, as the pixels outside it are always provided. This process is repeated for each dataset, meaning that two inpainting models were trained, one for each dataset.

At inference time, different to a normal SD inference pipeline, two extra inputs must be given: an image, on top of which the lesion will be inpainted, and a mask, with the designated region to inpaint. The diffusion process will be carried out as explained in Section 2.3.5, with the only difference being the conditional added by the two new inputs. Although a text prompt was given at training time, because it is the same for all samples, the same input prompt must be given during inference time, as described in Section 2.2.2.

### 2.5. Resources Management

Having three large models loaded at the same time, namely the denoising Unet, CLIP, and the VAE encoder, and enabling the gradient tracking for two of them during training, can represent a dramatic increase of GPU and processor resources. There exist techniques and frameworks to reduce this computing demand and fit the training requirements in a GPU memory of circa 20 GB with an efficient batch size of 256.

We used mixed precision using the *fp16* arithmetic, and the revision model specifically for that precision. When training in the (Ampere) A30 or A40 GPU, we activated the bf16 precision, with no improvement in the time or apparent quality of the training. Additionally, we used a lighter version of the AdamW optimizer, the 8-bit AdamW optimizer by Dettmers et al. [35], included in the *Bitsnadbytes* lightweight CUDA wrapper. Additionally, because our three models use attention layers, we used the *Xformers* efficient memory usage for transformers, which speeds up the training time and decreases the GPU usage.

To achieve a 256 batch size in one single GPU, we used gradient accumulation, a technique that consists of computing the gradient for a mini-batch without updating the model variables, for a set number of times, summing the gradients. In our case, using a mini-batch size of 16 and 16 gradient accumulation steps, the resulting accumulated batch size is 256. This technique, however, increased the overall training time.

Gradient checkpointing is another technique to reduce GPU memory usage. In this case, the CPU processors are used to release some GPU memory at the expenses of a slower training time. Gradient checkpointing saves strategically selected activations throughout the computational graph of the model tensors, so only a fraction of the activations must be re-computed for the gradients. A final memory reduction can be achieved by setting the optimizer gradients to *None* instead of zero after the weight updates have been completed. This will in general have a lower memory footprint and can modestly improve performance. Most of these techniques can be implemented directly using the *Hugging Face Accelerate* library and framework for distributed training and resources management.

## 3. Results

### 3.1. Healthy Mammogram Generation

As an initial experiment, we trained an unconditional diffusion model with the Hologic dataset using the same text prompt for all images: “a mammogram”. The evolution of the diffusion process as the training steps progress is shown in Figure 12. It can be seen that from the first epoch the generated image has essentially no signs of residual Gaussian noise, although the synthetic image does not resemble a mammogram. This implies that diffusion models pretrained on natural images have learned to denoise images and that the new task is to learn a new concept by finding its representation in the data distribution of the model. We can also notice that in three epochs the model has learned the significant characteristics of a mammogram and can generate realistic images. In the following epochs, the model focuses on improving smaller details on the image, such as the edges of the breast and the details of the breast parenchyma.

#### 3.1.1. Conditional Models

Training examples of the two separate conditional models using prompt text are shown in Figure 13 for the OMI-H dataset and Figure 14 for the VinDr dataset. We observe that the fine-tuning technique allows the generation of meaningful images from epoch one. For the Hologic example, we can observe that, as the training process increases, the mammogram reduces its shape in accordance to the area described in the prompt text. Moreover, it can be noticed that our models differentiate the overall intensity appearance of mammograms, which is different between Hologic and Siemens systems.

Thanks to the combined fine-tuning of the CLIP text encoder and the UNet weights, our conditional models can learn the anatomical structure and form of a mammogram, and can also push the generated image in the direction of the text prompt semantics as the training process increases.

#### 3.1.2. Joint OMI-H and VinDr Model: Concept Extrapolation

Besides allowing us to select the vendor type of the generated mammogram, the combination of both datasets permitted us to extrapolate the characteristics of one dataset into the other. This means that, e.g., the breast density of the generated Hologic mammograms can be controlled, even though this information was not available in the Hologic dataset. Figure 15 shows how the generated mammogram matches the text prompt characteristics as the training process advances, reducing the area and increasing the breast density.

#### 3.1.3. Guidance Scale: Quantitative Assessment

All of the synthetic examples above, despite being logged during training time, involve an inference pipeline. As explained in Section 2.3.5, there are two main hyperparameters that have to be tuned during inference.

First, the denoising steps, *T*, must be set. In our case, because we leveraged the DPM-solver of Lu et al. [32], 24 timesteps were enough for properly denoising the images. This usually means an average time of 2 s for the denoising of one sample. In some cases, due to the increase of the guidance scale, the number of *T* steps must be increased to completely remove the noise. The longest generation samples that we ran used T=50, needing a maximum of 4 s to denoise.

The guidance scale, on the other hand, played a more crucial role in the quality and diversity of the generated images. Figure 16 shows the effect of the guidance scale on the image generation. We observe that a guidance scale of 1 does not suffice for a meaningful generation. This is a common behavior for stable diffusion pipelines, as the image must be pushed further in the prompt direction (see Figure 7). It can be seen that the increase in the guidance value not only generates a more meaningful image but also adjusts the characteristics of the mammogram to better match the text prompt. For example, at guidance 2, the mammogram still presents low breast density, contrary to the text prompt description. In the following 3 and 4 guidance values, the breast density increases, as does the overall quality of the image.

Nevertheless, there exists a trade-off between prompt fidelity and generation diversity. If the guidance scale is high, the generated images diversity decreases, creating a similar behavior to the mode collapse suffered by GANs. To quantitatively assess this phenomenon, we computed the MS-SSIM metric for different guidance scale values. The MS-SSIM (Multi-Scale Structural Similarity Index) is usually used to assess the generation diversity of geenrative models. MS-SSIM is an extension of the traditional SSIM metric and measures the similarity between two images based on luminance, contrast, and structure.

The mean and standard deviation of the MS-SSIM values among 20 images at different guidance values using the same text prompt were computed and are shown in Table 3. The experiment was repeated for both vendors and the combined model. It can be seen that, overall, the higher the guidance value the lower the generation diversity, as the MS-SSIM value decreases. This suggests that the value of the guidance scale must be carefully selected, as a very low value will generate low quality images but with high diversity. Conversely, a high guidance value (above 6) will generate a mammogram more faithful to the prompt description but with low diversity. We attest that the optimum guidance scale will depend on the model, so empirical experiments using the MS-SSIM metric are encouraged. Nevertheless, the experiments we performed on all three models suggest that a guidance scale of 4 will suffice for a meaningful and diverse generation.

#### 3.1.4. Radiological Assessment

Visual assessment was performed with the radiological evaluation of 53 synthetic images by a radiologist. The experiment consisted of asking a radiologist with 30 years of experience to rate the mammograms on a scale of 0–4, where 0 means definitely synthetic image, 1 probably synthetic, 2 unsure, 3 probably real, and 4 definitely real image. The distribution of the mammograms had a 50/50 real–synthetic ratio. The results of the test are summarized as an ROC curve in Figure 17. The shape of the ROC curve bears resemblance to the random guess curve, suggesting that the radiologists cannot easily identify the difference between real and synthetic images. Moreover, the AUROC value obtained by the radiologist for this synthetic classification task was 0.49.

### 3.2. Lesion Generation

Initial results of the lesion generation pipeline show the possibility to inpaint mass-like lesions in any part of the mammogram. Visually, the lesions generated have a realistic appearance, although no radiological evaluation was carried out. As a preliminary experiment, we showed healthy Hologic mammograms with inpainted synthetic lesions to an in-house lesion classification CAD system trained with full field real Hologic mammograms containing masses and normal lesions. The heatmaps of three Explainability AI (XAI) methods were computed for those mammograms. The XAI interpretation methods applied were gradcam, saliency, and occlusion, and their respective heatmaps can be seen in Figure 18. The hypothesis is that when a synthetic mammogram is used as input the algorithm should highlight the synthetic lesion area, indicating that synthetic lesions have a similar pixel distribution to those present in real images.

### 3.3. MAM-E Graphical User Interfaces

We decided to build GUIs to make the pipelines of both tasks available to the public and easy to use. Our GUIs can run on remote servers and be accessible on the web thanks to *GradIO*, an open-source Python package for the rapid generation of the visual interface of ML models, by [36].

We developed five GUIs, one for each of our main diffusion pipelines. Two were designed for the conditional generation of mammograms of the original Siemens and Hologic datasets separately, with their own prompt characteristics. Similarly, one pipeline was created for the combination of both datasets, and it is presented as an example in Figure 1. In these three cases, the personalization options are set fixed and the user can only pick from the available options. Nevertheless, we added the option of a negative prompt, which allows the user to further personalize the generation.

The idea of the negative prompt is to specify some features that the user would like to be avoided. For instance, in the cases when a synthetic image presents a gray or white background, a negative prompt of “white background” or “no black background” has been shown to make the background black.

In the case of the inpainting task, the GUI has the option to upload the image that will be inpainted, although a default image is available. An interactive drawing brush is then activated, with which a lesion can be inpainted in any part of the mammogram, as shown in Figure 19.

Given that the pretrained weights are available in the *Hugging Face* personal repository of the first author, and that the code to run the GUI interface is publicly available in the GitHub repository of the same authorship, all five GUIs can be run with graphic cards of approximately 4 GB of GPU memory capacity.

## 4. Discussion

Our stable diffusion models show satisfactory results for the mammography generation task, capable of synthesizing visually realistic mammograms that are difficult to differentiate from real cases by a radiologist. Moreover, thanks to the text conditioning, we are capable of controlling the generation process and obtaining mammograms with specific features. Comparison with the work of Pinaya et al. [22] shows a similar visual quality of our synthetic images, with the main difference being that our conditional models control more than one image feature, namely vendor, view, breast density, and breast area. Additionally, regarding image diversity, even though the experiments of Pinaya et al. [22] do not completely correspond to ours, as we generated different scores for various guidance scales and datasets, we can attest from Table 3 a lower MS-SSIM score for our generated images compared with their 0.5356 MS-SSIM score, even for the largest guidance scales, meaning a higher image diversity.

Nevertheless, selecting the proper diffusion hyperparameters is challenging as, in some cases, the model may generate images with errors. Figure 20 shows some of the common generation issues faced by our SD models, in this case coming from the same text prompt at a guidance scale of 4.

These generation errors have different possible solutions, each of them with their drawbacks and limitations. For instance, the noise residuals in Figure 20a could be removed if the inference diffusion steps are increased, leading to longer generation time. The gray background issue in Figure 20b could be solved by using a *negative prompt*, which essentially specifies some features in the image that must be avoided, such as “white background” or “no black background”. The unsatisfied prompt description of Figure 20c and the nonsensical generation of Figure 20d could be solved by increasing the guidance scale at the expenses of the generation diversity, as previously discussed. Therefore, the selection of the optimal diffusion hyperparameter must be defined for each individual model empirically. For instance, as Table 3 shows, the optimal guidance scale value may not be the same across models. Moreover, in some cases, such as in the Siemens model, the effect of the guidance scale on the MS-SSIM value may not be significant and other metrics for image diversity must be computed for a better informed decision.

Another important outcome was the *concept extrapolation* property of the SD models. In this context, it means that semantic information among datasets can be shared between them during inference time. For instance, as seen in Figure 15, our model can generate a Siemens vendor mammogram controlling the breast area, even though this characteristic was not labeled in the original VinDr dataset. This property presents as a powerful additional characteristic of diffusion-based models as it increases the overall generation diversity of the model. The further exploration of this property is out of the scope of this work and will be addressed in future work.

The limitations of this work include the reduced resolution of the synthetic mammograms, which affects the use of our synthetic images on CAD system that require higher resolution, such as micro-calcification detection. The pixel depth was also reduced from its original 16 bits to 8 bits to match the pretrained model requirements, which leads to some information loss from the images and reduces the overall image contrast. Regarding model assessment, image synthesis is difficult to evaluate if not linked to a final application, such as a complete CAD pipeline implementation using synthetic images, but preliminary results are promising. Specific implementations include the use of the lesion inpainting pipeline for data augmentation on CAD systems where the number of lesion cases is much lower compared to normal cases, as happens in a screening population. However, this implementation may require the generation of other mammographic findings, such as microcalifications, architectural distortions, asymmetries, etc.

## 5. Conclusions

Conditional diffusion models are a promising generative model implementation to synthesize mammograms with specific characteristics and properties. In this work, we showed that fine-tuning a stable diffusion (SD) model pretrained on natural images with mammography data proves to be an effective strategy for the controlled generation of synthetic mammographic images. Additionally, we found that SD can be modified for the inpainting of synthetic lesions over healthy mammograms. The developed inpainting pipeline requires the modification of the input latent representation to include a mask to focus the generation process only in the masked region. Inference pipelines for these diffusion models were made accessible and ready-to-use through graphical user interfaces, and their weights and code can be found in the authors personal repositories.

We found initial evidence that synthetic images coming from our SD implementation could potentially be used for CAD systems in need of specific image characteristics, such as breast density, breast area, mammographic unit vendor, and the presence of lesions. A radiological assessment showed that the initial image quality can be compared with real mammograms, which indicates the prospective use of synthetic images for the training of radiologist or other educational usages. Moreover, explainability AI models allowed us to explore the behavior of a lesion classification model when processing our synthetic images, showing sensibility of the model to the synthesized lesions. Finally, we discovered a property of stable diffusion models we called *concept extrapolation*. When trained using datasets of populations with different labeled characteristics (breast density or area), stable diffusion allows controlling the generation process using labels not originally included in the training set of a specific dataset, augmenting the model’s generation customization.

Future work includes using the pretrained weights of the most recent SD models, such as the 768 × 768 resolution model or the incoming stable diffusion v3, which could allow higher resolution and diverse mammography generation. MAM-E could be used to generate population-based synthetic images for algorithm development and evaluation. Specifically, we plan to train complete CAD pipelines with and without synthetic images to analyze enhancement on the models performance when augmenting with synthetic images. In terms of assessment, we aim to perform a new radiological assessment including more radiologists to avoid bias and provide a more representative clinical opinion of the images. For the lesion inpainting task, we plan to add text conditionals to the training process to generate specific types of mammographic findings, such as microcalcifications and architectural distortions, with different biopsy diagnostics. Finally, we plan to explore the effects and benefits of *concept extrapolation* in more detail.

## Figures and Tables

**Figure 1 sensors-24-02076-f001:**
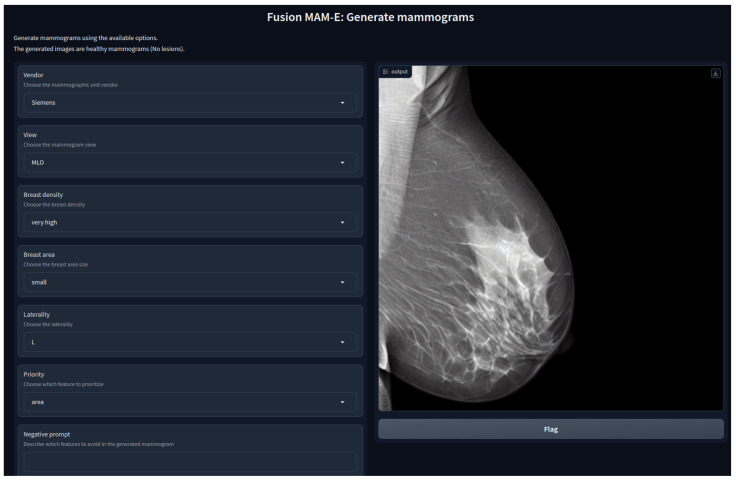
Graphical user interface of *MAM-E* for generation of synthetic healthy mammograms.

**Figure 2 sensors-24-02076-f002:**
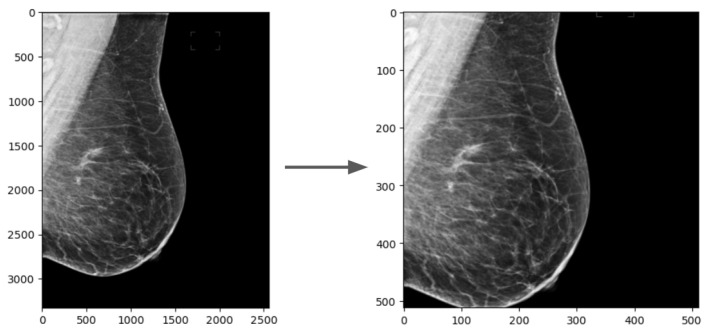
Resizing and cropping of an OMI-H mammogram. The same process was conducted for VinDr mammograms.

**Figure 3 sensors-24-02076-f003:**
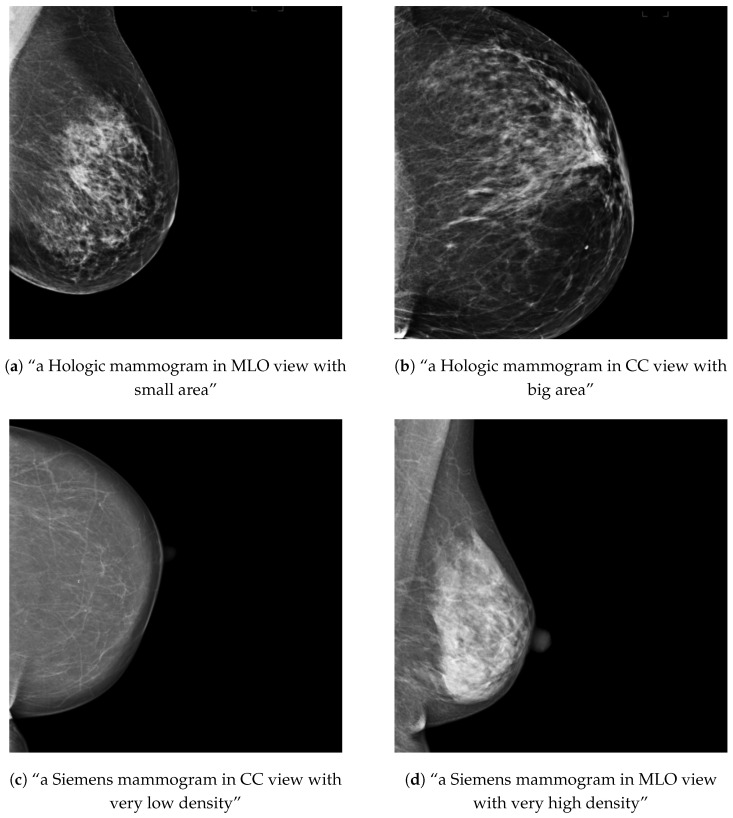
Examples of training mammograms (real) and their respective text prompts for OMI-H (**a**,**b**) and VinDr (**c**,**d**).

**Figure 4 sensors-24-02076-f004:**
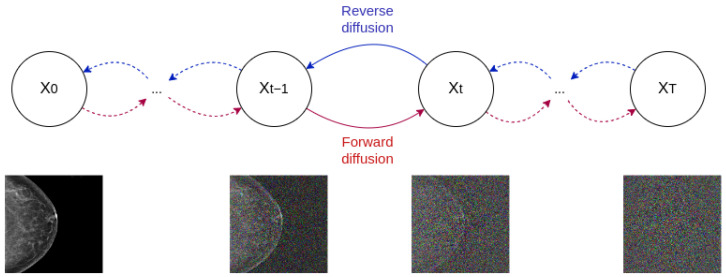
Forward and reverse diffusion process.

**Figure 5 sensors-24-02076-f005:**
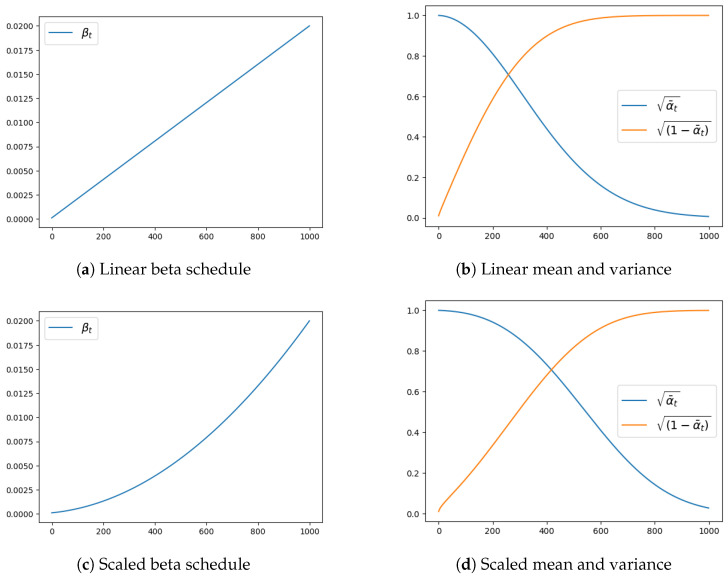
Linear and scaled beta schedulers (**left**) and their effects on the mean (blue) and variance (orange) of the noise sampling distributions (**right**).

**Figure 6 sensors-24-02076-f006:**
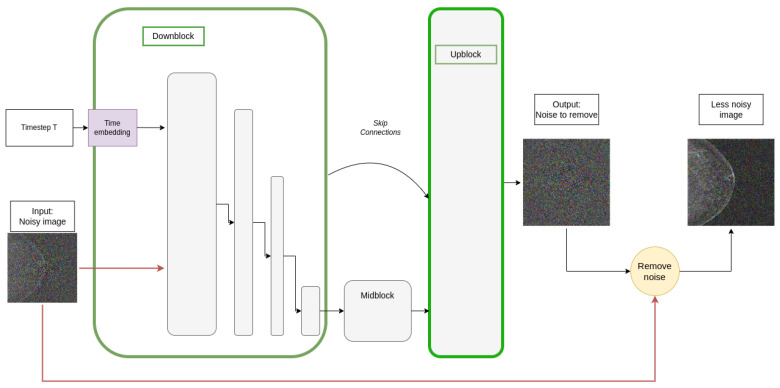
Reverse diffusion process using a denoising UNet. The upblock layers are a mirror of the downblock layers.

**Figure 7 sensors-24-02076-f007:**
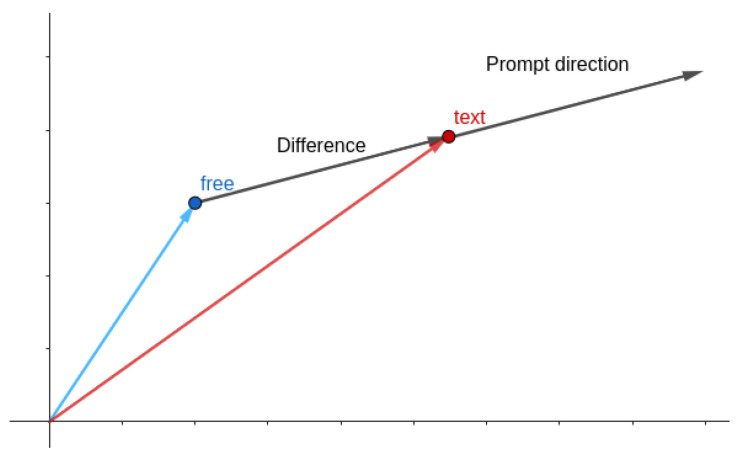
Classifier-free guidance geometrical interpretation. As the guidance scale increases, the image is pushed further in the prompt direction.

**Figure 8 sensors-24-02076-f008:**
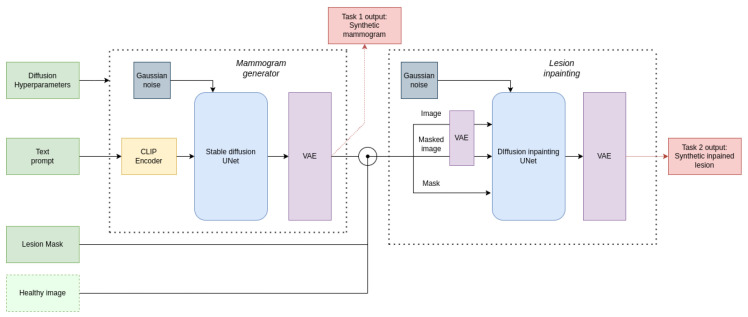
Overall MAM-E pipeline combining both full-field generation and lesion inpainting tasks. In dark green, the inputs needed for a full synthetic mammogram generation with lesion. In light green, the optional input for lesion inpainting on real images, instead of full-field synthetic images. In red, the outputs of each task.

**Figure 9 sensors-24-02076-f009:**
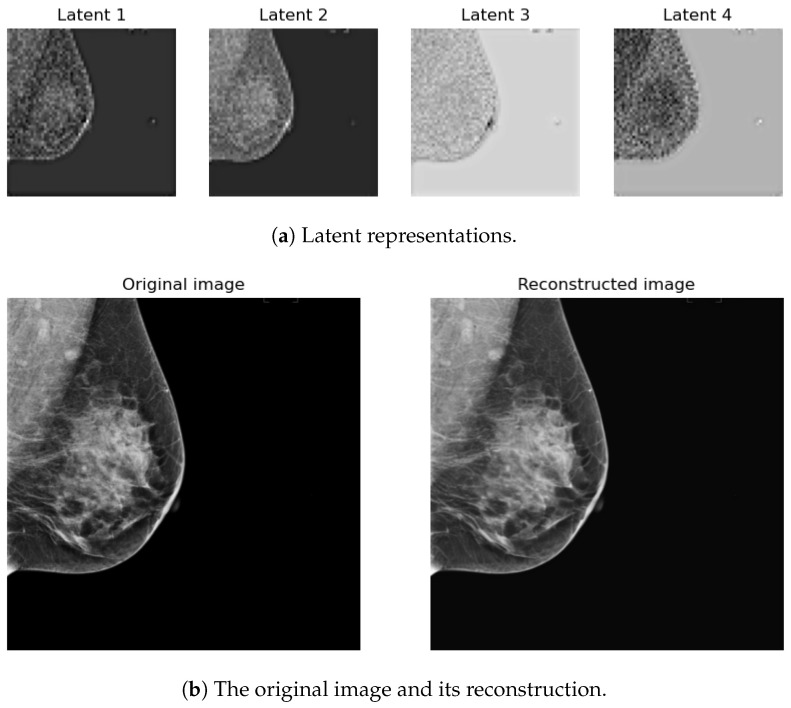
Example of the latent space representation of an image on the top and the original and reconstructed images on the bottom.

**Figure 10 sensors-24-02076-f010:**
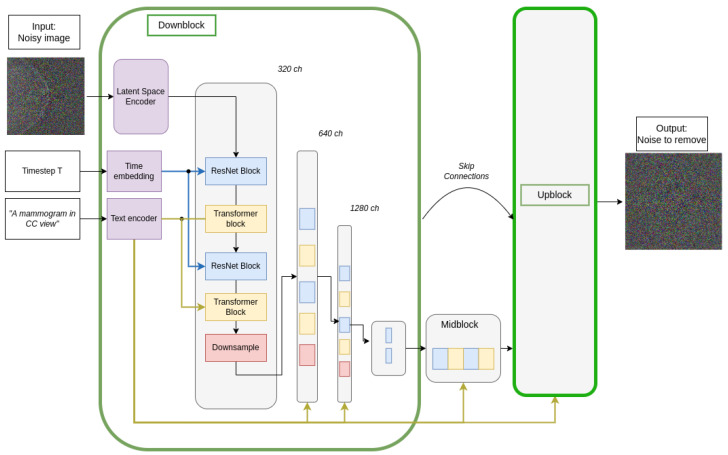
Denoising UNet architecture used for the reverse diffusion process. The upblock structure is a mirror of the downblock.

**Figure 11 sensors-24-02076-f011:**
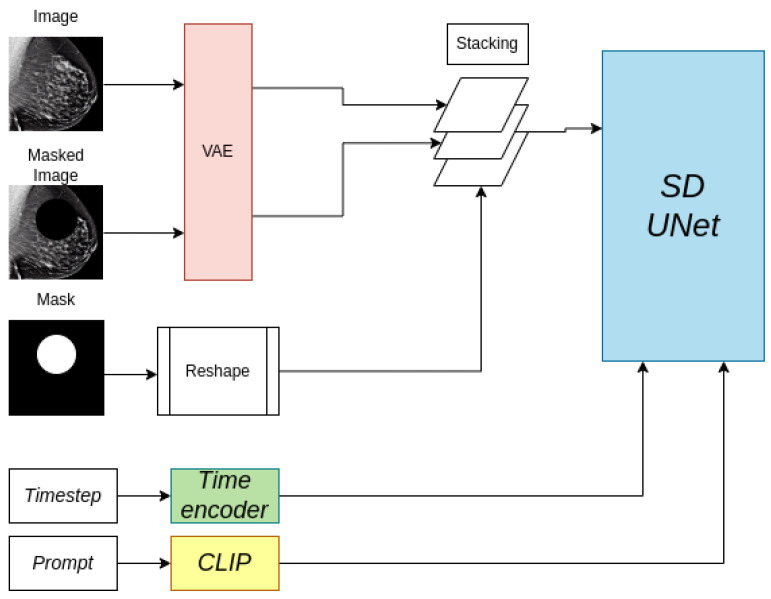
Inpainting training pipeline. The mask is reshaped to match the image size of the latent representations (64 × 64). The same UNet as in the Stable Diffusion pipeline is used.

**Figure 12 sensors-24-02076-f012:**
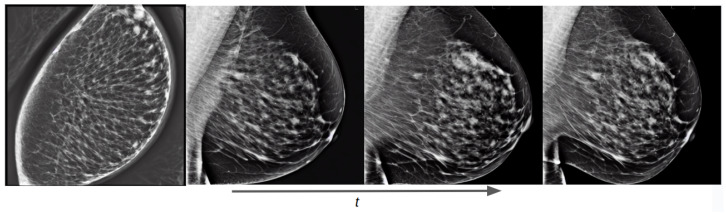
Training evolution of the diffusion process on an unconditional pretrained model at epochs 1, 3, 6, and 10.

**Figure 13 sensors-24-02076-f013:**
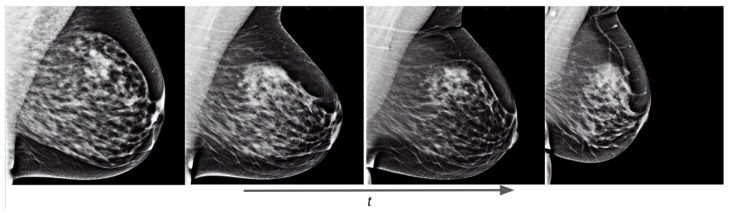
Training evolution of SD with Hologic images at epochs 1, 3, 6, and 10. The prompt is: “a mammogram in MLO view with small area”.

**Figure 14 sensors-24-02076-f014:**
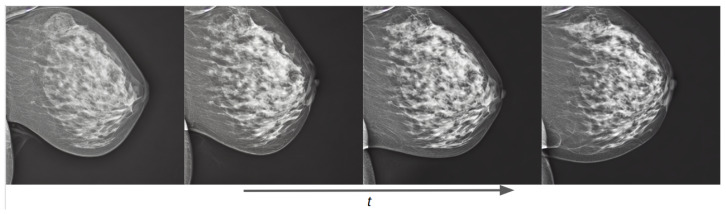
Training evolution of the diffusion process on a conditional pretrained model trained with Siemens images at epochs 1, 3, 6, and 10. The prompt is: “a mammogram in CC view with high density”.

**Figure 15 sensors-24-02076-f015:**
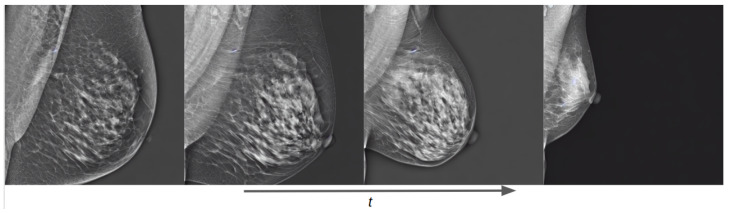
Training evolution of the diffusion process on a conditional pretrained model trained with both Siemens and Hologic images at epochs 1, 3, 7, and 40. The prompt is: “a siemens mammogram in MLO view with high density and small area”.

**Figure 16 sensors-24-02076-f016:**
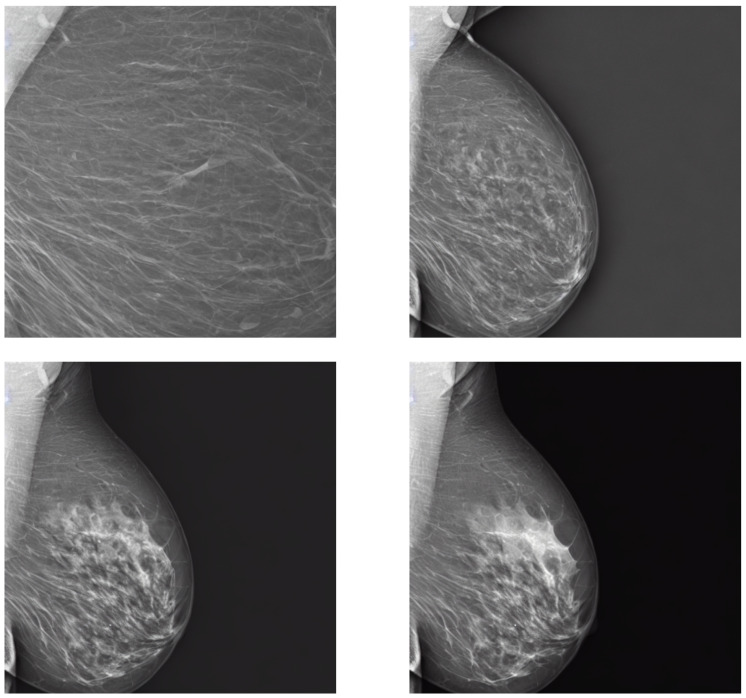
Guidance effect on the generation output. From upper-left to lower-right, the guidance varies in a range from 1 to 4. Prompt: “A siemens mammogram in MLO view with small area and very high density”.

**Figure 17 sensors-24-02076-f017:**
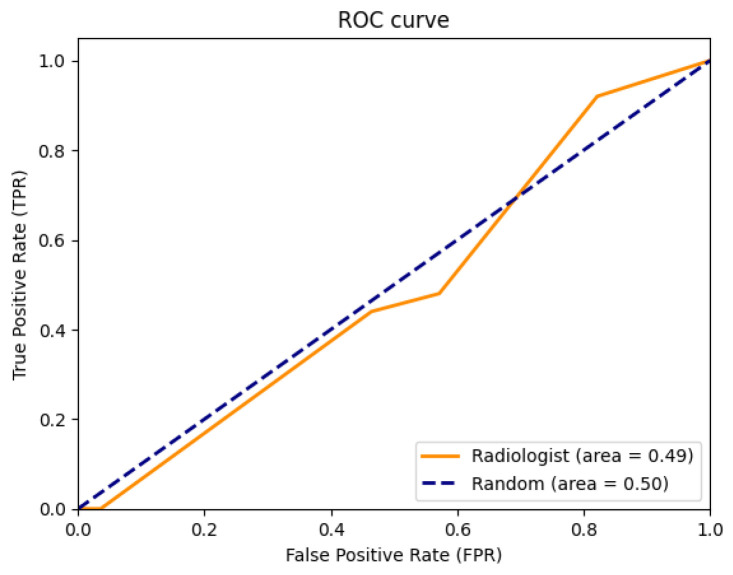
Receiver Operating Characteristic curve of radiological assessment.

**Figure 18 sensors-24-02076-f018:**
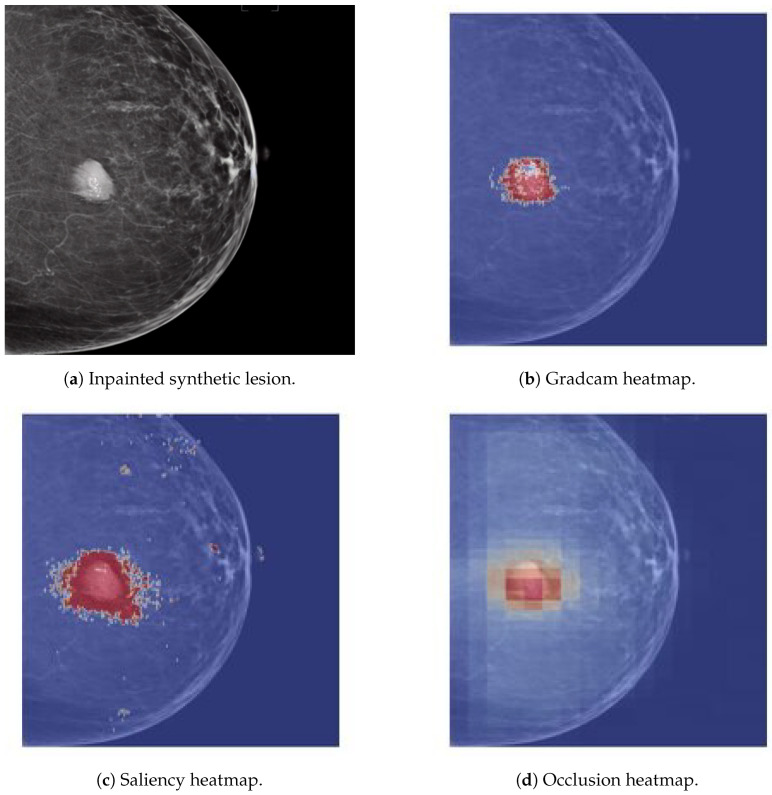
Explainability AI method heatmaps of synthetic lesion inpainted on real healthy mammograms.

**Figure 19 sensors-24-02076-f019:**
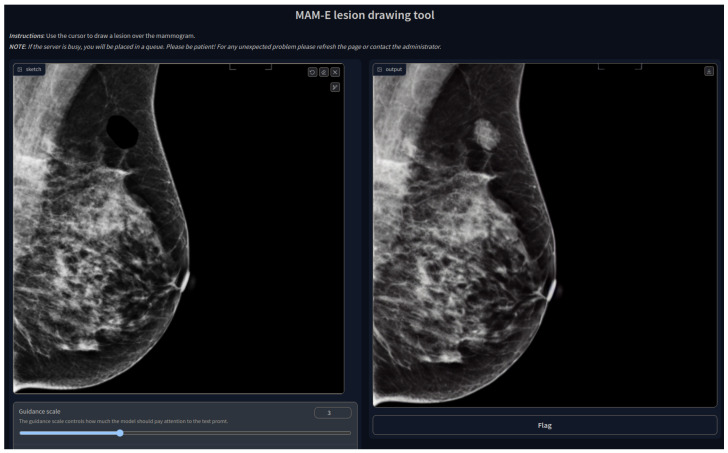
MAM-E lesion drawing tool.

**Figure 20 sensors-24-02076-f020:**
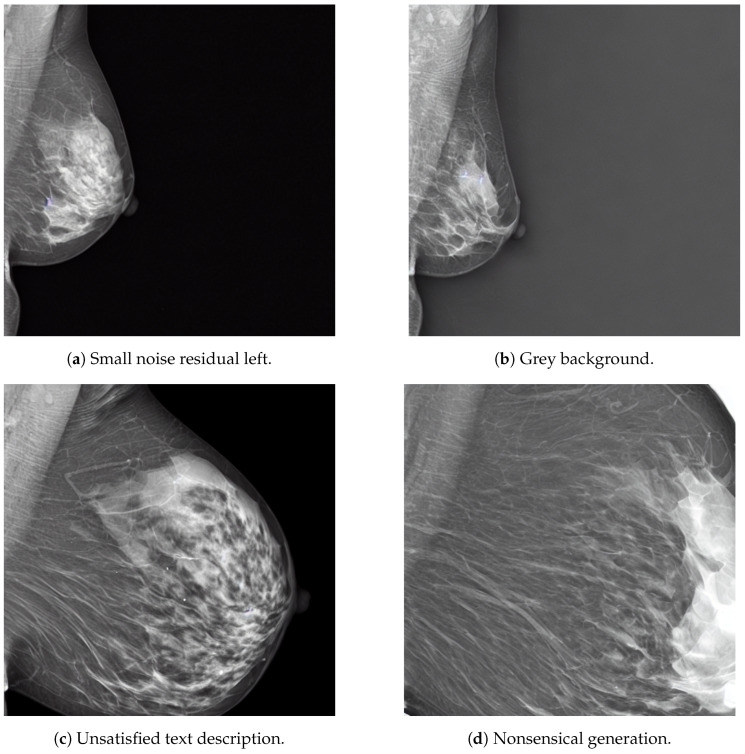
Examples of unsuccessful image generation of the combined dataset models coming from the same text prompt. The prompt was “A siemens mammogram in MLO view with small area and very high density” with a guidance scale of 4.

**Table 1 sensors-24-02076-t001:** Distribution of cases for both datasets.

	OMI-H	VinDr	Combined
Healthy	33,643	13,942	47,585
With lesion	6908	809	7717
Total	40,551	14,751	55,302

**Table 2 sensors-24-02076-t002:** Criteria for breast area size and breast density.

Breast area size	
Small	ratio < 0.4
Medium	0.4 < ratio < 0.6
Big	ratio > 0.6
Breast density	
Very low	Density A
Low	Density B
High	Density C
Very high	Density D

**Table 3 sensors-24-02076-t003:** Guidance scale effect on the MS-SSIM metric for the three SD models. The lower the MS-SSIM, the higher the image diversity.

	Hologic		Siemens		Fusion	
Guidance	Mean↓	STD	Mean↓	STD	Mean↓	STD
4	**0.29**	0.16	0.38	0.19	**0.37**	0.14
5	0.34	0.16	**0.36**	0.17	0.44	0.16
6	0.38	0.12	0.41	0.17	0.51	0.15
7	0.38	0.1	0.34	0.17	0.49	0.19
8	0.43	0.11	0.42	0.2	0.53	0.14
9	0.42	0.13	0.43	0.16	0.44	0.17
10	0.49	0.12	0.41	0.13	0.6	0.11
11	0.5	0.12	0.47	0.17	0.51	0.14
12	0.52	0.11	0.46	0.16	0.47	0.12
13	0.48	0.1	0.42	0.16	0.51	0.17
14	0.5	0.11	0.4	0.18	0.47	0.14

## Data Availability

Both OMI-DB [25] and VinDr [26] datasets, used in this work, are publicly available. The subset used in our experiments can be obtained from the dataset analysis found in the project repository and can also be obtained form the corresponding author upon reasonable request.

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
