# Peer review of "MAM-E: Mammographic Synthetic Image Generation with Diffusion Models"

_sensors, 2024, doi:10.3390/s24072076_

Round 1

Reviewer 1 Report

Comments and Suggestions for Authors

I have no idea how to suggest something different over here, when now-a-days, its just using available tools, modify those, and apply on the required case studies, and we try to publish it as a research paper! I mean, there is no harm in doing that.

However, did you relase your modified/u[dated codes and interface for public use? If so, please provide the links, if not; then I am curious to know why ? 

Also, what about comparisons ? There are only 3. What about you change the backbone model from U-Net and check whats under the hood/pipeline can offer. 

Reviewer 2 Report

Comments and Suggestions for Authors

The use of diffusion models for generating synthetic mammographic images represents a commendable attempt to address the issue of data scarcity in the field of medical imaging. I applaud the authors for their efforts in developing and open-sourcing these models. However, there are several critical concerns that need addressing.

Firstly, the paper lacks an essential introduction to mammography, crucial for contextualizing and motivating the research. This should include: 1) a discussion of the medical imaging applications where mammography is employed; 2) an explanation of the necessity for generating mammographic images; 3) a review of existing methods for mammographic image generation; and 4) an exploration of the unique challenges posed by using diffusion models in mammographic image generation.

Moreover, Section 2.3 requires substantial enhancement. Currently, it presents a basic overview of diffusion model techniques, with limited consideration of their design aspects specifically tailored to mammographic image generation. This lack of detail makes it difficult to understand the technical novelty of the paper. In addition, to improve clarity in this section, I suggest the following revisions: 1) Include a diagram illustrating the network structure of the diffusion models, as mentioned in the first paragraph of Section 2.3; 2) Provide a figure representing the overall network model used in this paper; 3) Offer background information on lesion inpainting, including its application scenarios.

Section 3 needs improvement. Specifically, 1) the metrics in Table 3 should be clearly defined, and the final Guidance Scale used in the paper should be presented; 2) to reduce bias in the radiological assessment, the involvement of more than one radiologist is recommended.

Lastly, the addition of a section dedicated to related work is crucial to appropriately position this paper within the broader research landscape.

Comments on the Quality of English Language

The Quality of English is good.

Round 2

Reviewer 1 Report

Comments and Suggestions for Authors

Unfortunately, I do not think it is still suitable for publishing. 

Author Response

We thank you for taking the time to review our work. Some final changes have been added after the comments of the academic editor.

Reviewer 2 Report

Comments and Suggestions for Authors

My comments have been addressed. Thanks for the efforts.

Author Response

Dear reviewer,

We thank you for your time reviewing our work and for the constructive comments and suggestions to improve it. After the comments of the academic editor, we have added some final changes to the introduction and conclusion, emphasizing the novelty of our work. 

Best regards,